# Women’s Views on Multifactorial Breast Cancer Risk Assessment and Risk-Stratified Screening: A Population-Based Survey from Four Provinces in Canada

**DOI:** 10.3390/jpm11020095

**Published:** 2021-02-02

**Authors:** Cynthia Mbuya-Bienge, Nora Pashayan, Jennifer D. Brooks, Michel Dorval, Jocelyne Chiquette, Laurence Eloy, Annie Turgeon, Laurence Lambert-Côté, Jean-Sébastien Paquette, Emmanuelle Lévesque, Julie Hagan, Meghan J. Walker, Julie Lapointe, Gratien Dalpé, Palmira Granados Moreno, Kristina Blackmore, Michael Wolfson, Yann Joly, Mireille Broeders, Bartha M. Knoppers, Anna M. Chiarelli, Jacques Simard, Hermann Nabi

**Affiliations:** 1CHU de Québec-Université Laval Research Center, Quebec City, QC G1V 4G2, Canada; cynthia.mbuya-bienge.1@ulaval.ca (C.M.-B.); Michel.Dorval@crchudequebec.ulaval.ca (M.D.); jocelyne.chiquette@crchudequebec.ulaval.ca (J.C.); Annie.Turgeon@crchudequebec.ulaval.ca (A.T.); Laurence.Lambert-Cote@crchudequebec.ulaval.ca (L.L.-C.); Julie.Lapointe@crchudequebec.ulaval.ca (J.L.); jacques.simard@crchudequebec.ulaval.ca (J.S.); 2Department of Social and Preventive Medicine, Faculty of Medicine, Université Laval, Quebec City, QC G1V 0A6, Canada; 3Department of Applied Health Research, Institute of Epidemiology and Healthcare, University College, London WC1E 6BT, UK; n.pashayan@ucl.ac.uk; 4Dalla Lana School of Public Health Science, University of Toronto, Toronto, ON M5S 1A1, Canada; jennifer.brooks@utoronto.ca (J.D.B.); Meghan.Walker@cancercare.on.ca (M.J.W.); Anna.Chiarelli@ontariohealth.ca (A.M.C.); 5Faculty of Pharmacy, Université Laval, Quebec City, QC G1V 4G2, Canada; 6CISSS de Chaudière-Appalaches Research Center, Lévis, QC G6V 3Z1, Canada; 7CHU de Québec—Université Laval, Quebec City, QC G1S 4L8, Canada; 8Département de Médecine Familiale et de Médecine d’Urgence, Université Laval, Quebec City, QC G1V 4G2, Canada; Jean-Sebastien.Paquette@fmed.ulaval.ca; 9Québec Cancer Program, Ministère de la Santé et des Services Sociaux, Quebec City, QC G1S 2M1, Canada; laurence.eloy@msss.gouv.qc.ca; 10Centre of Genomics and Policy, McGill University, Montreal, QC H3A 0G1, Canada; emmanuelle.levesque@mcgill.ca (E.L.); julie.hagan@mcgill.ca (J.H.); gratien.dalpe@mcgill.ca (G.D.); palmira.granadosmoreno@mcgill.ca (P.G.M.); yann.joly@mcgill.ca (Y.J.); bartha.knoppers@mcgill.ca (B.M.K.); 11Ontario Health, Cancer Care Ontario, Toronto, ON M5G 2L3, Canada; kristina.blackmore@ontariohealth.ca; 12School of Epidemiology and Public Health, University of Ottawa, Ottawa, ON K1G 5Z3, Canada; Michael.Wolfson@uOttawa.ca; 13Radboud Institute for Health Sciences, Radboud University Medical Center, 525 EZ Nijmegen, The Netherlands; Mireille.Broeders@radboudumc.nl; 14Dutch Expert Centre for Screening, 6538 SW Nijmegen, The Netherlands; 15Department of Molecular Medicine, Faculty of Medicine, Université Laval, Quebec City, QC G1V 4G2, Canada; 16Université Laval Cancer Research Center, Quebec City, QC G1R 3S3, Canada

**Keywords:** breast cancer, risk-stratified screening, population survey, precision prevention, women’s perspectives

## Abstract

Risk-stratified screening for breast cancer (BC) is increasingly considered as a promising approach. However, its implementation is challenging and needs to be acceptable to women. We examined Canadian women’s attitudes towards, comfort level about, and willingness to take part in BC risk-stratified screening. We conducted an online survey in women aged 30 to 69 years in four Canadian provinces. In total, 4293 women completed the questionnaire (response rate of 63%). The majority of women (63.5% to 72.8%) expressed favorable attitudes towards BC risk-stratified screening. Most women reported that they would be comfortable providing personal and genetic information for BC risk assessment (61.5% to 67.4%) and showed a willingness to have their BC risk assessed if offered (74.8%). Most women (85.9%) would also accept an increase in screening frequency if they were at higher risk, but fewer (49.3%) would accept a reduction in screening frequency if they were at lower risk. There were few differences by province; however, outcomes varied by age, education level, marital status, income, perceived risk, history of BC, prior mammography, and history of genetic test for BC (all *p* ≤ 0.01). Risk-based BC screening using multifactorial risk assessment appears to be acceptable to most women. This suggests that the implementation of this approach is likely to be well-supported by Canadian women.

## 1. Introduction

The Canadian Task Force on Preventive Health Care recommends that women aged 50–74 be screened with mammography every 2 to 3 years [1]. Still, there is a debate around the benefits and harms of breast cancer (BC) screening [2,3,4]. Risk-stratified BC screening, in which individual risk assessment based on multiple risk factors is used to tailor screening recommendations (e.g., more screening for women at higher risk and less screening for those at lower risk), has been proposed as an alternative to the current age-based approach [5,6,7]. Simulation models have shown this approach has the potential to increase the detection of breast cancers while decreasing false-positive outcomes and overdiagnosis, thereby overcoming the main limitations of today’s age-based screening programs [5,8].

Nonetheless, the implementation of this novel approach faces considerable organizational, social, ethical, and legal challenges. Obtaining real-world evidence and population’s engagement can help resolve these challenges [9,10,11]. There is also a consensus that for risk-based screening to be successfully implemented as part of a population-based screening program, it has to be accepted and supported by stakeholders, particularly women undergoing BC screening [11,12]. Burgeoning evidence from studies conducted in the United Kingdom (UK), Sweden, Netherlands, and Australia [12,13,14,15,16,17,18] indicates that women appear to welcome the prospect of risk-stratified BC screening.

The Canadian PERSPECTIVE I&I study (Personalized risk assessment for prevention and early detection of breast cancer: Integration and Implementation) is one of the major ongoing initiatives examining the potential of risk-stratified BC screening [3,19,20]. As part of this effort, the current study sought to examine attitudes towards, comfort level about, and willingness to take part in BC risk-stratified screening in Canadian women.

## 2. Materials and Methods

### 2.1. Study Design and Participants

A population-based, cross-sectional survey of women aged 30 to 69 years from the four largest provinces in Canada (Alberta, British Colombia, Ontario and Quebec) was conducted from in February 2019. Based on the age distribution of women in the four provinces in 2018 (https://www150.statcan.gc.ca/t1/tbl1/en/tv.action?pid=1710000501), we estimated a required sample size of 4268 women and applied quotas based on age and province. The Research Ethics Committees of the CHU de Québec-Université Laval and the McGill University approved the study (registration number: F9–42434).

### 2.2. Questionnaire Development

Based on previous studies [12,13,14,15,21,22], a multidisciplinary team of clinicians, epidemiologists, and social scientists developed the questionnaire. The structured online questionnaire developed in both French and English was piloted in a convenience sample of 100 women. This step did not reveal any problems; thus, no further changes were made. The questionnaire was then adapted to a web-based interface and administered by Ipsos Canada (https://www.ipsos.com/en), a survey firm with a panel of individuals from the general population in Canada who have previously consented to be contacted for research. Potential participants were emailed a link to the questionnaire.

### 2.3. Measures

#### 2.3.1. Outcomes Variables

Attitudes towards BC risk assessment and risk-based screening were measured using three questions: “What do you think of the idea of using information like age, cancers in your family, having children, lifestyle factors, breast density and weight to identify women who are at high, average or low risk of developing breast cancer?”; and “ What do you think of the idea of using results from genetic testing (i.e., analysis that checks for changes in your genetic makeup) to identify women who are at high, average or low risk of developing breast cancer?”; and “What do you think of the idea of changing how often women are invited for breast screening based on them being at high or low risk of developing breast cancer?” Participants were asked to use a 5-point Likert scale to report whether they viewed BC risk assessment and risk-stratified screening as a “very bad idea”, “bad idea”, “neither a good or a bad idea”, “good idea”, “very good idea”, “don’t know”, or “prefer not to answer”. As done previously [13,15], responses were dichotomized into either “good idea/very good idea” or “very bad/bad idea/neither a good or bad idea/don’t know/prefer not to answer”.

Being comfortable in providing information for BC risk assessment was measured using three questions: “How comfortable would you feel providing personal information (e.g., information regarding your lifestyle, personal and family medical history), so you can find out your breast cancer risk level?”, “How comfortable would you feel providing a small sample of blood or saliva for genetic testing (i.e., analysis that checks for features in your genetic makeup), so you can find out your breast cancer risk level?”, and “How comfortable would you feel having a mammogram to assess your breast density (amount of dense tissue compared to the amount of fatty tissue in the breast) so you can find out your breast cancer risk level?” Again, participants were given a 5-point Likert scale with choices ranging from “very comfortable” to “very uncomfortable.” The responses options were dichotomized into either “very comfortable/comfortable” or “very uncomfortable/uncomfortable/neither comfortable nor uncomfortable/prefer not to answer”.

Willingness to have BC risk assessed and tailored screening frequency was evaluated by the following four questions: “Would you be willing to have your breast cancer risk level assessed using information mentioned above? This would mean being told whether you are at average, lower than average, or higher than average risk of developing breast cancer”, “If your estimated level of breast cancer risk was higher than average, would you be willing to have your breast cancer screening more often than every 2 to 3 years?”, “If your estimated level of breast cancer risk was average or lower than average, would you be willing to have your breast cancer screening less often than every 2 to 3 years?”, and “If your estimated level of breast cancer risk was much lower than average, would you be willing not to be offered any breast screening?”. Responses were collected on a 5-point Likert scale with options dichotomized as “yes, definitely/yes, probably” vs. “no, definitely, not/no, probably, not/not sure/prefer not to answer”.

#### 2.3.2. Covariates

Sociodemographic variables were assessed for participants’ age, ethnicity, marital status, educational level, employment status, annual family income, and province of residence. Other covariates considered include perceived health status, personal history of BC, perceived lifetime susceptibility to BC, personal experience of mammography and genetic testing for BC, and family experience of genetic testing for BC.

### 2.4. Statistical Analysis

Descriptive statistics were used to present the characteristics of the study sample and to report proportions of participants according to each dichotomized outcome variables. Mutually adjusted logistic regression models were used to explore the associations between participants’ characteristics and the three outcomes of interest. We tested and found no sign of multicollinearity. Given the exploratory nature of the study and the correlation between the outcomes variables, we did not adjust for multiple comparisons [23]. Instead, we considered a more stringent p-value threshold (≤0.01) for statistical significance. All statistical analyses were performed using SAS version 9.4 (SAS Institute Inc., Cary, NC, USA).

## 3. Results

### 3.1. Sample Characteristics

Table 1 presents the characteristics of the study population. A total of 4293 women, with a mean age of 49.5 years, responded to the questionnaire (response rate of 63% based on the number of invitations sent). After excluding participants with missing data on one or several covariates (*n* = 74), our analytical sample included 4219 participants.

### 3.2. Attitudes Towards BC Risk Assessment and Risk-Stratified Screening

As shown in Panel A of Figure 1, the majority of women reported that it is a “good or very good idea” to use personal information (72.8%) and results from genetic tests (72.8%) to identify women who are at high, average, or low risk of developing BC, and to change how often women are invited for breast screening (63.5%) based on this information. Table 2 shows results of mutually adjusted logistic models exploring factors associated with attitudes towards risk assessment and risk-stratified screening. In general, women with a lower level of education and those who were single/never married were less likely to be in favour of risk-stratified BC screening (all *p*-values ≤ 0.01). In contrast, women with higher annual family income, higher perceived lifetime risk of BC, who previously had a mammogram or had a family history of genetic testing for BC were more likely to be in favour of risk-stratified screening (all *p*-values ≤ 0.01).

### 3.3. Being Comfortable in Providing Information for BC Risk Assessment

As shown in Panel B of Figure 1, most women reported to be “comfortable or very comfortable” in providing personal information (61.5%), a sample of blood or saliva for genetic testing (67.4%) and having a mammogram (66.9%) to find out their BC risk level. As shown in Table 3, older women, non-Caucasians, those with a high school diploma or less, single or never married, and those with a history of BC were less likely to report being “comfortable or very comfortable” in providing information necessary for BC risk assessment (all *p* values ≤ 0.01). In contrast, women with higher family income, good perceived health status, higher perceived lifetime risk of BC, previous experience of mammogram and those with a family history of genetic test for BC were more likely to report being “comfortable or very comfortable” in providing this information (all *p*-values ≤ 0.01).

### 3.4. Willingness to Have BC Risk Assessment and Tailored Screening Frequency

As shown in Panel C of Figure 1, a large majority of women reported that they “definitely or probably” would be willing to have their breast cancer risk level assessed (74.8%). A vast majority also stated that they “definitely or probably” would be willing to be screened more often than every 2 to 3 years if their estimated BC risk was higher than average (85.9%). Conversely, only about half of women said that they “definitely or probably” would be willing to be screened less frequently if their estimated BC risk was average or lower than average (49.3%). Further, most women stated that they would “definitely or probably not” be willing to not be offered any BC screening if their estimated risk was much lower than average (76.9%). As shown in Table 4, older women and, to some extent, those with lower education level were less willing to have their BC risk assessed and to have their screening more often if their risk is higher than average (all *p*-values ≤ 0.01). Older women were also more willing forego screening if their estimated BC risk was much lower than average (*p*-value < 0.01). In contrast, women with higher family income, higher perceived lifetime risk of BC, prior mammogram, and family history of genetic test for BC were more willing to have their BC risk assessed and to be screened more frequently if their estimated risk was higher than average (all *p*-values ≤ 0.01). Women with higher perceived lifetime risk of BC and prior mammogram were less willing to have less frequent or no screening if they were found to be at lower than average risk (all *p*-values ≤ 0.01). It was the opposite for those with personal or familial history of genetic test for BC (all *p* values ≤ 0.01).

## 4. Discussion

A large majority of Canadian women in the present study expressed favorable views of BC risk assessment and of a risk-stratified approach to BC screening. Most women reported being comfortable with providing personal and genetic information for BC risk assessment and showed a willingness to have their BC risk assessed if offered. Most women would then accept an increase in screening frequency if they were at higher than average risk, but fewer would accept a reduction if they were at lower risk. There were few differences by province of residence. It is worth mentioning that Canada has a universal healthcare system, but the healthcare organization is under provincial jurisdiction. Across Canada, BC screening programs are government-funded and are offered by different Provincial Health Services Authority agencies. Thus, BC screening programs might vary from one province to another. For instance, Ontario and British Columbia have a high-risk BC program while Alberta and Quebec do not have such a program. In high-risk programs, women have more frequent mammograms and use other diagnostic tests. However, attitudes were found to vary by age, education level, marital status, income, perceived risk of BC, personal history of BC, experience with mammography, and personal and family history of genetic test for BC (Appendix A).

These findings are consistent with those from previous quantitative studies conducted in the UK [13,14,15], Sweden [12], and Australia [16] showing that women are supportive of a BC screening approach based on multifactorial risk assessment. Overall, we found that 85.9% of Canadian women surveyed were willing to have more frequent breast screening if they were found to be at higher risk. Similar proportions were found in the UK (85.4%) [13] and Sweden (87%) [12]. Conversely, we found that only about half of them were willing to have less frequent screening if they were found to be at lower risk. Corresponding results from the UK (58%) [13] and Sweden (27%) [12] were also low but highly variable. We also asked participants if they would be willing to not be screened if they were found to be at very low risk. A majority would not accept this option. This suggests that despite the debates surrounding mammography screening, women are supportive of this public health measure, which might be considered as an acquired right [24]. Risk-stratified BC screening could be presented to the population as an opportunity to better balance the benefits and harms of mammography across the risk spectrum. Notably, close to one-quarter of women reported to be very uncomfortable or uncomfortable providing personal (25.6%) or genetic (21%) information for BC risk assessment. This could have implications for the implementation of risk-stratified screening. Although the underlying reasons for this reluctance remains to be investigated in future studies, fears and concerns about privacy and confidentiality might explain their discomfort [25].

Several participants’ characteristics were significantly associated with their views on risk-stratified screening. Specifically, those with a lower educational level and total family income exhibited less favorable views. This is consistent with findings from the UK [13] and Sweden [16], and implies that when considering implementing this approach, decision makers may need to find ways to engage women with low socioeconomic status who already tend to participate less in screening programs [26,27]. Having a higher perceived lifetime risk of BC was consistently associated with favorable attitudes and higher degree of comfort and willingness to undergo BC risk assessment and to have this risk assessment inform screening frequency. However, women who had this perception were also more likely to be against any reduction in screening frequency if found to be at lower risk of BC. This finding is consistent with the protection motivation theory which posits that high perceived risk of a disease is a strong predictor of adopting risk-reducing behaviours targeting that disease [28]. This could also explain the favourable views expressed by women with a personal history of BC or a personal or family history of genetic testing for BC, where these experiences might have influenced their perceived BC risk. Communication strategies to inform the general population need, therefore, to consider this influence of risk perception on the public adoption of risk-stratified screening.

This study has several strengths. To our best knowledge, this is the largest study on this topic. This allowed us to explore factors associated with women’s views of risk-stratified screening, information needed to determine implementation strategies. Further, the use of a web-based survey provided a cost-effective approach to include younger women aged 30 to 49 who are not currently eligible for screening in provincial programs. Limitations include the potential for selection bias with respect to those women who chose to participate and complete the study questionnaire. In addition, we included women from only four provinces, which limits the generalisability of the findings across Canada, even though we found little differences by province. However, several characteristics of our sample, including marital status, education, income, ethnicity, and self-rated health, are fairly comparable to those of the participants of the Statistics Canada’s Canadian Community Health Surveys [29,30].

## 5. Conclusions

The present study provides evidence suggesting that the implementation of risk-stratified BC screening is likely to be well-supported by Canadian women. Further studies are needed to understand why some women are reluctant to provide their personal and genetic information for BC risk assessment. Future studies should also look on how the interactions between different factors shape women’s views on stratified BC screening. Finally, additional studies are also needed to generate evidence on the acceptability of this approach by healthcare professionals, on the ways to communicate BC risk, and to assess healthcare organisational readiness and costs.

## Figures and Tables

**Figure 1 jpm-11-00095-f001:**
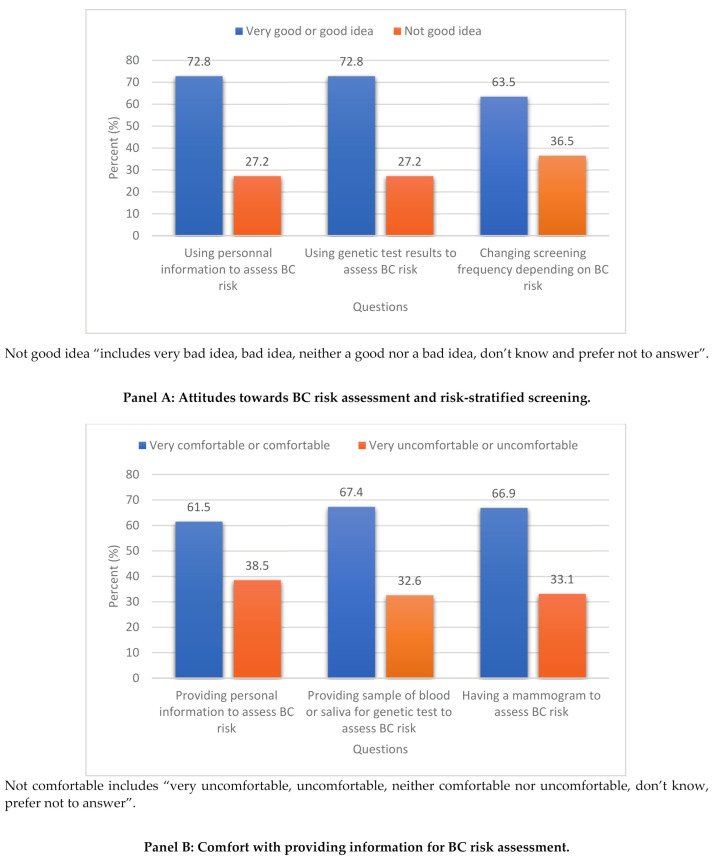
Attitudes, comfort level and willingness to take part in breast cancer risk-stratified screening for Canadian women.

**Table 1 jpm-11-00095-t001:** Characteristics of the study population.

Characteristics	Total (*n* = 4219)	% (95% CI)
Age groups		
30–39 years 40–49 years 50–59 years 60–69 years	1045 1046 1069 1059	24.8 (23.5–26.1) 24.8 (23.5–26.1) 25.3 (24.0–26.7) 25.1 (23.8–26.4)
Province		
Alberta British Colombia Ontario Quebec	1057 1061 1058 1043	25.0 (23.8–26.5) 25.2 (23.8–26.4) 25.1 (23.8–26.4) 24.7 (23.4–26.1)
Country of birth		
Canada Other	3565 654	84.5 (83.4–85.6) 15.5 (14.4–16.6)
Ethnicity		
Caucasian Others ^a^ Unknown ^b^	3410 681 128	80.8 (79.6–82.0) 16.1 (15.0–17.3) 3.1 (2.5–3.6)
Education level		
High school diploma or less	1170	27.7 (26.4–29.1)
Non-university certificate or post-secondary diploma	1832	43.4 (41.9–44.9)
University diploma	1217	28.9 (27.5–30.2)
Marital status		
Married or common law Formerly married (widowed/divorced/separated) Single, never married Prefer not to answer	2627 754 793 45	62.2 (60.8–63.7) 17.9 (16.7–19.1) 18.8 (17.6–20.0) 1.1 (0.8–1.4)
Employment status		
Working (full time or part time) Not working Retired Prefer not to answer	2502 841 827 49	59.3 (57.8–60.8) 19.9 (18.7–21.2) 19.6 (18.4–20.8) 1.2 (0.9–1.5)
Total family income		
Less than $20,000 $20,000–$39,999 $40,000–$59,999 $60,000–$79,999 $80,000 or more Don’t know/Prefer not to answer	325 621 731 588 1394 560	7.7 (6.9–8.6) 14.7 (13.7–15.8) 17.3 (16.2–18.5) 13.9 (12.9–15.0) 33.1 (31.6–34.5) 13.3 (12.3–14.3)
Perceived health status		
Good ^c^ Fair ^d^	3427 792	81.2 (80.0–82.4) 18.8 (17.6–20.0)
Ever had breast cancer		
Yes No	172 4047	4.1 (3.5–4.7) 95.9 (95.3–96.5)
Perceived lifetime risk of breast cancer		
Much lower or lower than others The same as others Much higher or higher than others Don’t know	1172 2076 682 289	27.8 (26.4–29.2) 49.2 (47.7–50.7) 16.2 (15.1–17.3) 6.8 (6.1–7.7)
Ever had a mammogram		
Yes No Don’t know/Prefer not to answer	2620 1561 38	62.1 (60.6–63.6) 37.0 (35.5–38.5) 0.9 (0.6–1.2)
Ever had a genetic test for breast cancer		
Yes No Don’t know/Prefer not to answer	260 3710 249	6.2 (5.5–6.9) 87.9 (86.9–88.9) 5.9 (5.2–6.7)
Family history of genetic test for breast cancer		
Yes No Don’t know/Prefer not to answer	483 2395 1341	11.4 (10.5–12.5) 56.8 (55.3–58.3) 31.8 (30.4–33.2)

^a^ Others include Aboriginal, Asian, Black, Latin American, and Arab. ^b^ Unknown includes the answers “don’t know” and “prefer not to answer”. ^c^ The category good includes the answers “excellent”, “very good” and “good”. ^d^ The category fair includes the answers “fair” and “poor”.

**Table 2 jpm-11-00095-t002:** Mutually adjusted logistic regression models for the associations between attitudes towards breast cancer risk assessment and risk-stratified screening with participants’ characteristics (*n* = 4219).

Questions: What Do You Think of the Idea of:	Using Personal Information * to Assess BC Risk?		Using Genetic Test Results to Assess BC Risk?		Changing Screening Frequency Depending of BC Risk?	
	Very Good; Good Idea vs. (Very Bad; Bad Idea; Neither a Good Nor Bad Idea; Don’t Know; Prefer Not to Answer)					
Characteristics	Adjusted OR, (95% CI)	*p*-Value	Adjusted OR, (95% CI)	*p*-Value	Adjusted OR, (95% CI)	*p*-Value
Age groups						
30–39 years 40–49 years 50–59 years 60–69 years	1.00 0.98 (0.80–1.21) 1.04 (0.81–1.32) 1.08 (0.81–1.45)	0.8638 0.7691 0.6102	1.00 0.90 (0.73–1.12) 0.82 (0.64–1.04) 0.78 (0.58–1.04)	0.3483 0.1022 0.0913	1.00 1.04 (0.86–1.26) 0.90 (0.72–1.13) 0.86 (0.66–1.12)	0.7014 0.3544 0.2507
Province						
Ontario Alberta British Colombia Quebec	1.00 1.08 (0.88–1.33) 1.00 (0.81–1.22) 0.82 (0.67–1.00)	0.4622 0.9752 0.0585	1.00 0.97 (0.79–1.19) 0.92 (0.75–1.12) 1.01 (0.82–1.24)	0.7589 0.3895 0.9421	1.00 1.08 (0.90–1.29) 0.99 (0.83–1.19) 0.94 (0.78–1.14)	0.4284 0.9490 0.5448
Country of birth						
Canada Other	1.00 0.91 (0.73–1.13)	0.3803	1.00 0.95 (0.77–1.09)	0.6733	1.00 1.03 (0.84–1.26)	0.7735
Ethnicity						
Caucasian Others Don’t know/Prefer not to answer	1.00 0.80 (0.64–0.99) 0.51 (0.35–0.76)	0.0415 0.0008	1.00 0.88 (0.71–1.09) 0.59 (0.40–0.88)	0.2267 0.0084	1.00 1.02 (0.84–1.24) 0.58 (0.40–0.83)	0.8783 0.0035
Education level						
University diploma Non-university certificate or post-secondary diploma High school diploma or less	1.00 0.72 (0.60–0.87) 0.52 (0.42–0.64)	0.0006 <0.0001	1.00 0.89 (0.74–1.07) 0.75 (0.61–0.92)	0.2176 0.0065	1.00 0.98 (0.83–1.15) 0.90 (0.75–1.16)	0.8127 0.2538
Marital status						
Married or common law Formerly married Single, never married Prefer not to answer	1.00 1.05 (0.85–1.30) 0.98 (0.80–1.20) 1.16 (0.57–2.36)	0.6393 0.8193 0.6836	1.00 1.03 (0.84–1.27) 0.91 (0.74–1.10) 1.24 (0.61–2.50)	0.7493 0.3240 0.5519	1.00 0.96 (0.79–1.16) 0.76 (0.63–0.91) 0.93 (0.48–1.79)	0.6545 0.0027 0.8196
Employment status						
Working Not working Retired Prefer not to answer	1.00 0.98 (0.80–1.19) 1.22 (0.95–1.57) 0.36 (0.19–0.69)	0.8226 0.1154 0.0020	1.00 1.01 (0.83–1.23) 1.16 (0.91–1.47) 0.45 (0.24–0.84)	0.9187 0.2244 0.0120	1.00 1.07 (0.89–1.29) 0.93 (0.75–1.15) 0.81 (0.44–1.50)	0.4462 0.4883 0.5037
Total family income						
Less than $20,000 $20,000–$39,999 $40,000–$59,999 $60,000–$79,999 $80,000 or more Don’t know/Prefer not to answer	1.00 1.05 (0.77–1.43) 1.22 (0.89–1.66) 1.31 (0.94–1.85) 1.40 (1.01–1.93) 1.06 (0.76–1.48)	0.7497 0.2217 0.1156 0.0444 0.7514	1.00 1.21 (0.89–1.63) 1.51 (1.11–2.05) 1.81 (1.29–2.53) 1.68 (1.22–2.30) 1.24 (0.89–1.72)	0.2192 0.0089 0.0006 0.0014 0.1993	1.00 1.07 (0.80–1.42) 1.14 (0.85–1.52) 1.40 (1.02–1.92) 1.25 (0.92–1.68) 1.11 (0.81–1.51)	0.6587 0.3886 0.0352 0.1500 0.5284
Perceived health status						
Poor Good	1.00 1.21 (1.00–1.45)	0.0479	1.00 1.18 (0.98–1.41)	0.0836	1.00 1.00 (0.84–1.18)	0.9698
Ever had breast cancer						
No Yes	1.00 0.61 (0.42–0.87)	0.0075	1.00 0.62 (0.43–0.90)	0.0127	1.00 0.93 (0.66–1.32)	0.6908
Perceived lifetime risk of breast cancer						
Much lower or lower than others The same as others Much higher or higher than others Don’t know	1.00 1.09 (0.93–1.30) 2.10 (1.62–2.71) 0.40 (0.30–0.53)	0.2945 <0.0001 <0.0001	1.00 1.11 (0.94–1.31) 1.82 (1.42–2.34) 0.44 (0.34–0.58)	0.2106 <0.0001 <0.0001	1.00 0.91 (0.78–1.06) 1.42 (1.14–1.77) 0.49 (0.38–0.65)	0.2013 0.0017 <0.0001
Ever had a mammogram						
No Yes Don’t know/Prefer not to answer	1.00 1.32 (1.09–1.60) 0.43 (0.21–0.90)	0.0043 0.0242	1.00 1.31 (1.08–1.58) 0.36 (0.18–0.76)	0.0057 0.0069	1.00 1.12 (0.94–1.34) 0.39 (0.19–0.80)	0.2026 0.0099
Ever had a genetic test for breast cancer						
No Yes Don’t know/Prefer not to answer	1.00 0.78 (0.56–1.08) 0.56 (0.42–0.77)	0.1388 0.0002	1.00 1.31 (0.92–1.87) 0.64 (0.48–0.86)	0.1388 0.0033	1.00 1.02 (0.75–1.39) 1.03 (0.77–1.37)	0.9014 0.8576
Family history of genetic test for breast cancer						
No Yes Don’t know/Prefer not to answer	1.00 1.27 (0.98–1.64) 1.16 (0.98–1.38)	0.0701 0.0788	1.00 1.45 (1.11–1.89) 1.21 (1.02–1.42)	0.0064 0.0270	1.00 1.45 (1.14–1.83) 1.12 (0.96–1.30)	0.0021 0.1504

* Personal information like age, cancers in your family, having children, lifestyle factors, breast density and weight.

**Table 3 jpm-11-00095-t003:** Mutually adjusted logistic regression models for the associations between comfort with providing information for breast cancer risk assessment and participants ‘characteristics (*n* = 4219).

Questions: How Comfortable Would You Feel	Providing Personal Information * to Assess Breast Cancer Risk?		Providing Sample of Blood or Saliva for Genetic Test to Assess Breast Cancer Risk?		Having a Mammogram to Assess Breast Cancer Risk?	
	Very Comfortable; Comfortable vs. (Very Uncomfortable; Uncomfortable; Neither Comfortable nor Uncomfortable; Don’t Know; Prefer Not to Answer)					
Characteristics	Adjusted OR, (95% CI)	*p*-Value	Adjusted OR, (95% CI)	*p*-Value	Adjusted OR, (95% CI)	*p*-Value
Age groups						
30–39 years 40–49 years 50–59 years 60–69 years	1.00 0.77 (0.63–0.93) 0.76 (0.61–0.95) 0.63 (0.48–0.82)	0.0080 0.0177 0.0007	1.00 0.79 (0.65–0.96) 0.79 (0.63–0.99) 0.77 (0.58–1.00)	0.2190 0.0115 0.8861	1.00 0.88 (0.72–1.08) 0.73 (0.58–0.93) 1.02 (0.76–1.37)	0.2190 0.0115 0.8861
Province						
Ontario Alberta British Colombia Quebec	1.00 0.96 (0.80–1.15) 1.03 (0.85–1.23) 1.19 (0.98–1.44)	0.6611 0.7808 0.0755	1.00 0.90 (0.75–1.09) 1.00 (0.83–1.20) 1.49 (1.23–1.81)	0.2740 0.9919 <0.0001	1.00 1.03 (0.84–1.26) 1.25 (1.03–1.52) 0.97 (0.79–1.19)	0.7734 0.0315 0.7592
Country of birth						
Canada Other	1.00 0.94 (0.77–1.16)	0.5683	1.00 0.85 (0.69–1.04)	0.1086	1.00 0.85 (0.68–1.06)	0.1384
Ethnicity						
Caucasian Others Don’t know/Prefer not to answer	1.00 0.78 (0.64–0.96) 0.34 (0.23–0.51)	0.0194 <0.0001	1.00 1.09 (0.89–1.33) 0.51 (0.35–0.75)	0.4322 0.0007	1.00 0.92 (0.74–1.13) 0.56 (0.38–0.84)	0.4181 0.0052
Education level						
University diploma Non-university certificate or post-secondary diploma High school diploma or less	1.00 0.90 (0.76–1.06) 0.72 (0.60–0.87)	0.2071 0.0008	1.00 0.96 (0.82–1.14) 0.77 (0.64–0.93)	0.6579 0.0752	1.00 0.98 (0.82–1.17) 0.81 (0.66–0.99)	0.8233 0.0424
Marital status						
Married or common law Formerly married Single, never married Prefer not to answer	1.00 1.15 (0.95–1.39) 0.78 (0.65–0.94) 0.46 (0.22–0.98)	0.1611 0.0084 0.0438	1.00 1.10 (0.91–1.34) 0.77 (0.64–0.93) 0.48 (0.24–0.97)	0.3308 0.0063 0.0418	1.00 0.95 (0.77–1.17) 1.01 (0.83–1.23) 0.59 (0.29–1.17)	0.6258 0.8940 0.1316
Employment status						
Working Not working Retired Prefer not to answer	1.00 1.15 (0.95–1.38) 1.25 (0.87–1.56) 0.71 (0.37–1.37)	0.1434 0.0442 0.3025	1.00 1.02 (0.85–1.23) 0.99 (0.80–1.24) 0.68 (0.36–1.30)	0.8312 0.9482 0.2451	1.00 1.07 (0.88–1.30) 1.07 (0.83–1.37) 0.80 (0.42–1.55)	0.5076 0.6057 0.5112
Total family income						
Less than $20,000 $20,000–$39,999 $40,000–$59,999 $60,000–$79,999 $80,000 or more Don’t know/Prefer not to answer	1.00 1.21 (0.90–1.62) 1.16 (0.87–1.56) 1.27 (0.93–1.74) 1.59 (1.17–2.15) 0.72 (0.53–0.99)	0.2053 0.3129 0.1405 0.0029 0.0411	1.00 1.23 (0.92–1.66) 1.16 (0.86–1.56) 1.30 (0.95–1.80) 1.44 (1.06–1.96) 0.76 (0.55–1.04)	0.1685 0.3200 0.1034 0.0190 0.0823	1.00 1.21 (0.90–1.65) 1.62 (1.19–2.21) 1.67 (1.20–2.33) 2.27 (1.64–3.13) 1.36 (0.97–1.89)	0.2094 0.0022 0.0026 <0.0001 0.0733
Perceived health status						
Fair Good	1.00 1.15 (0.97–1.37)	0.1030	1.00 0.98 (0.82–1.17)	0.8499	1.00 1.40 (1.17–1.68)	0.0003
Ever had breast cancer						
No Yes	1.00 0.66 (0.47–0.94)	0.0195	1.00 0.65 (0.46–0.93)	0.0178	1.00 0.77 (0.52–1.14)	0.1863
Perceived lifetime risk of breast cancer						
Much lower or lower than others The same as others Much higher or higher than others Don’t know	1.00 0.85 (0.73–1.00) 1.51 (1.21–1.88) 0.41 (0.31–0.54)	0.0463 0.0003 <0.0001	1.00 0.99 (0.85–1.16) 1.58 (1.26–1.98) 0.44 (0.33–0.58)	0.9195 <0.0001 <0.0001	1.00 1.27 (1.08–1.50) 2.09 (1.64–2.66) 0.73 (0.52–0.98)	0.0047 <0.0001 0.0277
Ever had a mammogram						
No Yes Don’t know/Prefer not to answer	1.00 1.38 (1.16–1.65) 0.22 (0.09–0.54)	0.0003 0.0011	1.00 1.31 (1.10–1.57) 0.32 (0.14–0.71)	0.0030 0.0051	1.00 4.36 (3.60–5.27) 0.38 (0.16–0.90)	<0.0001 0.0277
Ever had a genetic test for breast cancer						
No Yes Don’t know/Prefer not to answer	1.00 0.96 (0.71–1.31) 0.77 (0.58–1.03)	0.8013 0.0801	1.00 1.08 (0.78–1.48) 0.69 (0.52–0.93)	0.6446 0.0136	1.00 0.62 (0.45–0.86) 0.64 (0.47–0.87)	0.0037 0.0045
Family history of genetic test for breast cancer						
No Yes Don’t know/Prefer not to answer	1.00 1.38 (1.09–1.75) 1.03 (0.89–1.20)	0.0073 0.6979	1.00 1.60 (1.25–2.04) 1.17 (1.00–1.36)	0.0002 0.0457	1.00 1.50 (1.16–1.93) 0.91 (0.78–1.08)	0.0020 0.2767

* Personal information like age, cancers in your family, having children, lifestyle factors, breast density and weight.

**Table 4 jpm-11-00095-t004:** Mutually adjusted logistic regression models for the associations between willingness to have breast cancer risk assessment and to modify breast screening and participants’ characteristics (*n* = 4219).

Questions: Would You be Willing	To Have Your BC Risk Level Assessed?		To Have Your BC Screening More Often if BC Risk Higher Than Average?		To Have Your BC Screening Less Often if BC Average or Lower Than Average?		Not to Be Offered any BC Screening if BC Risk Much Lower Than Average?	
	Yes, Definitely; Yes, Probably vs. (No, Probably Not; No, Definitely Not; Don’t Know; Prefer Not to Answer)							
Characteristics	Adjusted OR, (95% CI)	*p*-Value	Adjusted OR, (95% CI)	*p*-Value	Adjusted OR, (95% CI)	*p*-Value	Adjusted OR, (95% CI)	*p*-Value
Age groups								
30–39 years	1.00		1.00		1.00		1.00	
40–49 years	0.81 (0.65–1.00)	0.0522	0.91 (0.70–1.18)	0.4685	1.02 (0.85–1.29)	0.8568	1.02 (0.82–1.27)	0.8684
50–59 years	0.66 (0.51–0.85)	0.0013	0.59 (0.44–0.81)	0.0009	1.04 (0.78–1.30)	0.7235	1.45 (1.13–1.86)	0.0034
60–69 years	0.63 (0.46–0.85)	0.0024	0.59 (0.40–0.86)	0.0066	1.01 (0.92–1.31)	0.9552	1.40 (1.04–1.89)	0.0257
Province								
Ontario	1.00		1.00		1.00		1.00	
Alberta	1.00 (0.82–1.24)	0.9338	1.17 (0.90–1.53)	0.2362	1.10 (0.92–1.31)	0.3010	1.12 (0.91–1.37)	0.3034
British Colombia	1.05 (0.86–1.30)	0.6265	1.18 (0.91–1.53)	0.2186	0.88 (0.74–1.05)	0.1630	0.98 (0.79–1.20)	0.8083
Quebec	1.08 (0.87–1.33)	0.4976	1.20 (0.92–1.57)	0.1715	0.82 (0.69–0.98)	0.0301	1.00 (0.81–1.24)	0.9738
Country of birth								
Canada	1.00		1.00		1.00		1.00	
Other	0.89 (0.71–1.12)	0.3186	0.95 (0.72–1.26)	0.7150	1.07 (0.88–1.30)	0.5108	1.30 (1.04–1.62)	0.0221
Ethnicity								
Caucasian	1.00		1.00		1.00		1.00	
Others	1.08 (0.86–1.36)	0.4988	0.83 (0.63–1.08)	0.1650	1.18 (0.97–1.43)	0.0906	1.29 (1.04–1.60)	0.0231
Don’t know/Prefer not to answer	0.41 (0.28–0.60)	<0.0001	0.42 (0.27–0.65)	0.0001	0.71 (0.50–1.04)	0.0823	0.73 (0.44–1.20)	0.2168
Education level								
University diploma	1.00		1.00		1.00		1.00	
Non-university certificate or post-secondary diploma	0.96 (0.79–1.15)	0.6426	0.92 (0.73–1.18)	0.5210	0.99 (0.85–1.16)	0.9151	1.07 (0.89–1.28)	0.4952
High school diploma or less	0.80 (0.65–0.99)	0.0423	0.71 (0.54–0.93)	0.0117	0.99 (0.84–1.19)	0.9476	1.00 (0.80–1.24)	0.9737
Marital status								
Married or common law	1.00		1.00		1.00		1.00	
Formerly married	1.15 (0.93–1.43)	0.2071	1.03 (0.78–1.35)	0.8565	0.98 (0.82–1.17)	0.7968	1.00 (0.81–1.24)	0.9889
Single, never married	0.91 (0.74–1.12)	0.3731	0.82 (0.64–1.15)	0.1083	0.83 (0.70–1.00)	0.0447	0.96 (0.78–1.19)	0.7058
Prefer not to answer	0.97 (0.49–1.94)	0.9297	2.34 (0.89–6.17)	0.0866	0.71 (0.37–1.38)	0.3090	0.68 (0.28–1.69)	0.4112
Employment status								
Working	1.00		1.00		1.00		1.00	
Not working	1.15 (0.93–1.43)	0.1760	1.08 (0.84–1.39)	0.5330	1.01 (0.85–1.21)	0.8741	0.84 (0.68–1.03)	0.0927
Retired	1.10 (0.86–1.40)	0.4664	1.34 (0.96–1.86)	0.0852	0.87 (0.70–1.06)	0.1697	0.84 (0.66–1.08)	0.1740
Prefer not to answer	0.89 (0.47–1.42)	0.7295	0.65 (0.31–1.34)	0.2394	1.21 (0.65–2.26)	0.5427	0.82 (0.38–1.79)	0.6220
Total family income								
Less than $20,000	1.00		1.00		1.00		1.00	
$20,000–$39,999	1.15 (0.83–1.59)	0.4030	1.40 (0.97–2.01)	0.0719	1.34 (1.01–1.78)	0.0442	0.92 (0.67–1.27)	0.6213
$40,000–$59,999	1.09 (0.79–1.51)	0.5889	1.85 (1.26–2.70)	0.0015	1.47 (1.11–1.95)	0.0079	0.81 (0.58–1.12)	0.2021
$60,000–$79,999	1.31 (0.56–1.13)	0.1280	1.92 (1.27–2.91)	0.0020	1.06 (0.78–1.44)	0.7083	0.76 (0.53–1.08)	0.1205
$80,000 or more	1.44 (1.03–2.02)	0.0343	2.34 (1.57–3.48)	<0.0001	1.09 (0.81–1.46)	0.5648	0.75 (0.53–1.05)	0.0906
Don’t know/Prefer not to answer	0.82 (0.58–1.16)	0.2665	1.45 (0.97–2.16)	0.0674	0.99 (0.73–1.35)	0.9561	0.57 (0.40–0.82)	0.0026
Perceived health status								
Fair	1.00		1.00		1.00		1.00	
Good	1.12 (0.93–1.37)	0.2365	1.08 (0.85–1.37)	0.5313	1.11 (0.94–1.30)	0.2378	0.96 (0.79–1.17)	0.6814
Ever had breast cancer								
No	1.00		1.00		1.00		1.00	
Yes	0.75 (0.50–1.11)	0.1527	1.00 (0.59–1.71)	0.9934	0.90 (0.65–1.25)	0.5362	1.17 (0.81–1.71)	0.4077
Perceived lifetime risk of breast cancer								
Much lower or lower than others	1.00		1.00		1.00		1.00	
The same as others	1.12 (0.94–1.33)	0.2004	1.37 (1.10–1.70)	0.0046	0.74 (0.64–0.85)	<0.0001	0.58 (0.49–0.69)	<0.0001
Much higher or higher than others	2.04 (1.56–2.66)	<0.0001	2.31 (1.62–3.28)	<0.0001	0.65 (0.53–0.80)	<0.0001	0.47 (0.37–0.60)	<0.0001
Don’t know	0.37 (0.28–0.49)	<0.0001	0.49 (0.36–0.68)	<0.0001	0.44 (0.34–0.59)	<0.0001	0.55 (0.40–0.77)	0.0004
Ever had a mammogram								
No	1.00		1.00		1.00		1.00	
Yes	1.91 (1.57–2.33)	<0.0001	2.82 (2.20–3.61)	<0.0001	0.91 (0.77–1.08)	0.2818	0.65 (0.53–0.79)	<0.0001
Don’t know/Prefer not to answer	0.29 (0.13–0.63)	0.0016	0.24 (0.12–0.52)	0.0002	0.36 (0.17–0.79)	0.0111	0.59 (0.24–1.46)	0.2519
Ever had a genetic test for breast cancer								
No	1.00		1.00		1.00		1.00	
Yes	0.92 (0.64–1.32)	0.6396	0.62 (0.41–0.94)	0.0240	1.13 (0.85–1.51)	0.3926	1.58 (1.16–2.16)	0.0040
Don’t know/Prefer not to answer	0.71 (0.52–0.97)	0.0315	0.40 (0.28–0.57)	<0.0001	0.92 (0.70–1.22)	0.5619	1.11 (0.79–1.55)	0.5551
Family history of genetic test for breast cancer								
No	1.00		1.00		1.00		1.00	
Yes	1.72 (1.29–2.29)	0.0003	1.34 (0.95–1.88)	0.0913	1.34 (1.08–1.66)	0.0086	1.26 (0.99–1.62)	0.0630
Don’t know/Prefer not to answer	1.13 (0.95–1.34)	0.1551	1.50 (1.19–1.87)	0.0004	1.14 (0.99–1.32)	0.0753	0.94 (0.79–1.12)	0.4697

## Data Availability

The study questionnaire is available upon request to the corresponding author.

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
