# Peer review of "Women’s Views on Multifactorial Breast Cancer Risk Assessment and Risk-Stratified Screening: A Population-Based Survey from Four Provinces in Canada"

_jpm, 2021, doi:10.3390/jpm11020095_

Round 1

Reviewer 1 Report

The study is a population-based survey from Canada, for evaluating the attitute of women about risk-stratified screening mammogram program.

It is interesting subject, and the result and presentation is very clear.

I have some minor conscerns.

First, authors include the province as a factor for analysis.

I wonder why you thought the province would be a factor how a woman could answer the quastions.

Are there any differences between privnces, eg. cultural difference, socioeconimic difference or health care system?

Second, it would be better if the authors woud add a paragraph about the health system.

The result showed that women are willing to increase the frequency of screening mammogram, but have some resistance to reduce it. I guess the health care system could be a very important base of the resul. Therefore, who would pay the fee for screening mammogram, the goverment, private insurance, or individual person, should be explained.

As a minor problem, please check the typo in Panel C, the last column. ' Not to be offeref-'

Reviewer 2 Report

This is a well written paper describing the risk stratification assessment for breast cancer screening in Canadian women from 4 provinces.

1) Introduction can include initially what is the current guidelines for breast cancer screening

2) Have the authors considered full adherence biases ? If so how would you address this bias

Author Response

Pleas see attachment
